# Legumes Regulate Symbiosis with Rhizobia via Their Innate Immune System

**DOI:** 10.3390/ijms24032800

**Published:** 2023-02-01

**Authors:** Estelle B. Grundy, Peter M. Gresshoff, Huanan Su, Brett J. Ferguson

**Affiliations:** 1Integrative Legume Research Group, School of Agriculture and Food Sciences, The University of Queensland, Brisbane 4072, Australia; 2National Navel Orange Engineering Research Centre, College of Life Science, Gannan Normal University, Ganzhou 341000, China

**Keywords:** legumes, nodulation, pathogen, plant immune system, rhizobia, symbiosis

## Abstract

Plant roots are constantly exposed to a diverse microbiota of pathogens and mutualistic partners. The host’s immune system is an essential component for its survival, enabling it to monitor nearby microbes for potential threats and respond with a defence response when required. Current research suggests that the plant immune system has also been employed in the legume-rhizobia symbiosis as a means of monitoring different rhizobia strains and that successful rhizobia have evolved to overcome this system to infect the roots and initiate nodulation. With clear implications for host-specificity, the immune system has the potential to be an important target for engineering versatile crops for effective nodulation in the field. However, current knowledge of the interacting components governing this pathway is limited, and further research is required to build on what is currently known to improve our understanding. This review provides a general overview of the plant immune system’s role in nodulation. With a focus on the cycles of microbe-associated molecular pattern-triggered immunity (MTI) and effector-triggered immunity (ETI), we highlight key molecular players and recent findings while addressing the current knowledge gaps in this area.

## 1. Introduction

### 1.1. The Importance of Legumes and Nodulation

Legumes represent a major and diverse group of flowering plants, including some of the most important crops for food, feed, and pasture economies globally [1,2,3]. The significance of legumes to global agriculture may be attributed to their unique influence on the nitrogen cycle, coupled with exceptional nutritional qualities and consistent high yields.

Nitrogen is the key macronutrient for plant growth and development and an important factor in sustaining productive farms. Modern agricultural practices are heavily reliant on the production of synthetic nitrogen fertiliser, which is often used in excess to maintain high yields in the face of growing demand [4,5]. However, N fertiliser usage in agriculture has large-scale negative consequences for our environmental systems, including pollution of neighboring ecosystems such as the eutrophication of waterways and the release of harmful greenhouse gases such as nitrous oxide (NO_x_) [6]. Furthermore, the production of synthetic nitrogen fertiliser is expensive, largely inefficient, and does not provide a reliable long-term solution for maintaining global food security [1,4,7]. Alternatively, legumes can minimise their reliance on nitrogen by entering into a unique mutualistic symbiosis with compatible bacterial microsymbionts, which are broadly referred to as rhizobia. These rhizobia species convert atmospheric nitrogen gas (N_2_) into a usable nitrogen form (i.e., ammonia (NH_3_)) for plant growth, development, and eventual crop yield. This relies on a complex and highly specific process termed “nodulation” which is characterised by the production of novel organs on the plant’s roots, termed nodules. These organs house the microsymbionts, providing them with suitable conditions for biological nitrogen fixation. In exchange for the nitrogen source, plant hosts provide carbohydrates, as well as other macro- and micronutrients, for the enveloped rhizobia [8].

To initiate this process, plant roots exude secondary metabolites termed flavonoids into the surrounding soil to attract their rhizobial partners [9]. Rhizobia perceive the flavonoids, which induce the NodD transcriptional activator, and in response, rhizobial produce lipo-chito-oligosaccharides (LCOs), known as nodulation factors (NF; Figure 1), via *NodABC* gene transcription [8,10,11,12]. NFs are structurally defined by an oligomeric chitin backbone of beta-1,4-linked N-acetylglucosamine residues, containing various substitutions on nonreducing and reducing ends depending on the species. For example, substitutions often include the acylation of different kinds of fatty acids or sulfation [13,14]. Once secreted, NFs are perceived by plant-encoded Nod Factor Receptors 1 and 5 (NFR1 and NFR5) in *Lotus japonicus* and *Glycine max/Glycine soja* (soybean) [15,16,17,18], Nod Factor Perception (NFP) and LYSIN MOTIF DOMAIN-CONTAINING RECEPTOR-LIKE KINASE 3 (LYK3) for *Medicago truncatula* [19,20], and SYM2A and SYM10 in *Pisum sativum* (pea) [21,22]. This perception is a crucial point in determining host-specificity between the two partners [16]. If incompatible, the plant will upregulate host defences, restricting the rhizobia from invasion [23]. In compatible interactions, NFs have been shown to stimulate root hair deformation, the expression of nodulin genes, and the formation of nodule primordia, thereby facilitating both the infection process and nodule formation [14,24,25].

Plants further regulate this interaction via a process termed the autoregulation of nodulation (AON) pathway [26]. This pathway functions by restricting the nodule number to avoid excessive nodule formation and therefore conserve plant resources when required. This process restricts rhizobial infection, as it does the plant immune system in incompatible plant-microbe interactions. However, none of the key factors so far identified structurally represent defence proteins involved in microbe-associated molecular pattern-triggered immunity (MTI) and/or effector-triggered immunity (ETI). Therefore, this pathway is distinct from the infection mechanisms explored here.

**Figure 1 ijms-24-02800-f001:**
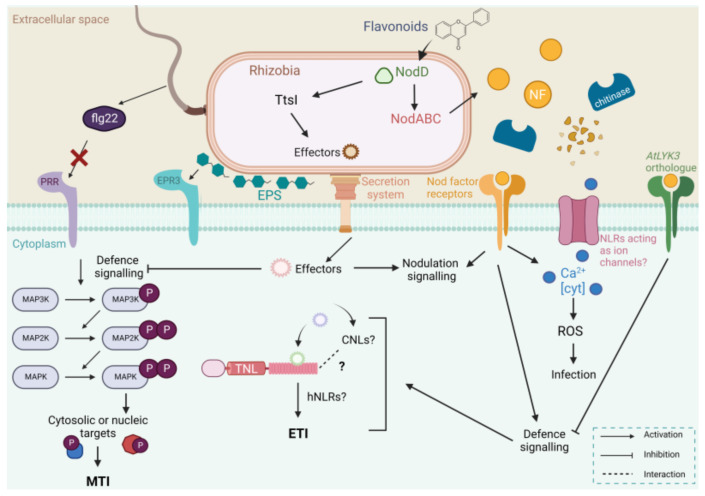
A general model for the interplay of nodulation and defence signalling mechanisms induced by rhizobial infection at the root hair cell. To attract rhizobial partners, legume roots exude flavonoids into the rhizosphere, which then bind to the rhizobial transcriptional activator NodD. In response, rhizobia produce Nod factors (NFs) via *NodABC* gene transcription. Secreted NFs are recognised by their cognate plant Nod factor receptors, activating symbiotic signalling to promote nodulation. However, NFs have also been shown to induce a small defence response upon recognition, including the production of reactive oxygen species (ROS) and the upregulation of defence genes. NFs also attenuate defences in the nonlegume *Arabidopsis* via LYSIN MOTIF-CONTAINING RECEPTOR-LIKE KINASE 3 (AtLYK3). Chitinases are implicated in this host-specific recognition via NF degradation activity. Additionally, host plants also contain pattern recognition receptor (PRR) membrane proteins to perceive bacterial microbial-associated molecular patterns (MAMPs), triggering a mitogen-activated protein kinase (MAPK) signalling cascade that leads to an associated defence response and MAMP-triggered immunity (MTI). These include FLAGELLINSENSITIVE2 (FLS2) and exopolysaccharide receptor 3 (Epr3), which function to perceive flagellin and exopolysaccharides (EPS), respectively. However, certain rhizobial flagellins, such as that of *Sinorhizobium meliloti,* lack the flagellin 22 (flg22) epitope for perception [27,28]. NodD also induces the TtsI transcriptional activator, which initiates type III secretion system (T3SS) transcription. Certain rhizobia secrete nodulation outer protein (Nop) proteins via secretion systems (T3SS, type IV secretion system, and type VI secretion system) to attenuate plant defences and promote infection and establishment. Resistant plants have evolved to produce nucleotide binding site-leucine-rich repeat receptor (NLR) proteins to recognise these Nops. Recognition activates defence signalling once again, leading to the inhibition of infection and effector-triggered immunity (ETI). Plant-pathogen NLRs form calcium-permeable channels that induce ROS and cell death but have not yet been identified in nodulation. MAP2K—MAPK kinase; MAP3K—MAP2K kinase; TNL—Toll/interleukin 1 (TIR) NLR; CNL—coiled-coil (CC)-NLR; hNLRs—helper NLRs; Ca^2+^—calcium ions; Cyt—cytosolic and P—phosphate group. Created with BioRender.com.

### 1.2. The Plant Innate Immune System in Nodulation

Aside from mutualistic rhizobia, legumes can also interact with a wide variety of other microbes at the root-soil interface, including beneficial and pathogenic fungi, parasitic nematodes, and benign endophytes [29]. It is therefore crucial to the plant’s survival to discriminate between beneficial partners and detrimental pathogens [30]. As a result, many plants have developed a two-tiered innate immune system equipped with highly specific detection mechanisms to recognise a range of microbes and respond accordingly [31]. There is considerable evidence to suggest that this immune system is essential in determining the success of the legume-rhizobia symbiosis. While rhizobia can serve as beneficial partners by supplying nitrogen to the plant, certain strains can act as saprophytes or even parasites, where they receive carbohydrates from the plant without fixing nitrogen [32,33,34]. The innate immune system is thought to regulate the selection of potential symbionts by identifying and inhibiting the invasion of certain rhizobia species. Indeed, plant hosts have been shown to regulate partner selection in the presence of diverse microbial communities [35]. Accordingly, competitive rhizobia are thought to have evolved to evade and modulate these host defences for successful infection and establishment [35,36]. Here, such as plant-pathogen interactions, host-specificity relies on the genetic compatibility of the legume host and the invading species [37]. In this review, we summarise what is currently known about the impact and roles of early immune responses involving MTI and ETI in establishing legume nodulation.

After these initial infection processes, the immune system will continue to govern the development of the symbiosis. This occurs during rhizobial internalisation and symbiosome formation within the plant’s nodules. Here, multiple factors are required to mediate immunity in order to promote symbiosis between *M. truncatula* and *S. meliloti* [38]. For example, *M. truncatula* genes *DEFECTIVE IN NITROGEN FIXATION 2 (MtDNF2)* and *SYMBIOTIC CYSTEINE-RICH RECEPTOR-LIKE KINASE* (*MtSymCRK)* are proposed to function in preventing plant defences during this stage of symbiosis. Mutants of DNF2 and SymCRK developed nodules with elevated defence gene expression, phenolic compounds, and reduced bacterial viability, indicating the induction of an immune response in the absence of these two genes [39,40,41]. For further details on the plant immune system during rhizobial internalisation and later stages post initial infection, please see Berrabah et al. [38].

## 2. Initial Rhizobial Infection Results in a Transient and Local Plant Defence Response

Current research supports the view that early rhizobial infection is accompanied by an initial defence response by the plant, which is then suppressed by the rhizobia to allow nodulation to occur [42]. A wide range of transcriptomic analyses have demonstrated transient induction of host defence-related genes during early stages of rhizobial infection of soybean, *L. japonicus,* and *M. truncatula* [42,43,44,45]. Specifically, in *L. japonicus*, the compatible *Mesorhizobium loti* treatment resulted in induced defence responses such as an increase in mitogen-activated protein kinase (MAPK) phosphorylation [43]. MAPKs function in signalling cascades to relay extracellular signals to the nucleus for an associated response (Figure 1). This often involves a MAPKKK (MAP3K), which activates a MAPKK (MAP2K), which in turn activates a MAPK [46]. This MAPK activation is similar to what is expected following treatment with pathogenic bacteria [28]. Other defences found to be upregulated in these datasets include *Resistance (R)* genes, pathogenesis-related (PR) proteins such as chitinases and peroxidases, phytoalexin biosynthesis genes, and genes involved in cell wall modification [42,43,44,45].

However, this induced defence response is transient, with subsequent downregulation of defence genes. For example, in soybean RNA-seq data, we recorded defence-related genes exhibiting enhanced expression at 12 and 24 hours post-inoculation (hpi), which declined by 48 hpi [42]. Similarly, in the *M. truncatula* transcriptomic analysis, many of the annotated defence genes were upregulated at 1 hpi and subsequently downregulated at 6–12 hpi [44]. These transcriptomic findings led to a hypothetic model where suppression of transient immune responses is essential for rhizobia invasion and establishment [23,36,42]. The downregulation of defence genes, such as *R* genes, is required for nodulation to occur, as demonstrated by microRNA regulation of *R* gene targets which suppresses *R* gene expression and promotes nodulation [47]. Constitutive expression of defence genes is energetically costly and can result in abnormal developmental phenotypes such as stunted growth, autoimmunity, and inhibition of nodulation [48,49]. Therefore, suppressing defence responses quickly after they are induced would be beneficial to symbiosis.

In contrast, subsequent transcriptomic analysis of *L. japonicus* revealed dissimilar host responses to symbiotic and pathogenic infection [50]. The authors found no transient defence responses occurring in response to compatible rhizobia, contradicting previous findings and hypotheses. Kelly et al. [50] suggested that the use of different genotypes for both the host and rhizobial strain, as well as different growing conditions, may account for the differences observed. This result has also been demonstrated in the legume *Aeschynomene evenia*, where the compatible rhizobia did not induce the expression of a defence-related gene [51]. Together, these findings suggest that the upregulation of defence-related genes by the host may not be a general response across all legume-rhizobia relationships and that certain genotypic variations may account for observed differences.

Despite inconsistencies in transcriptomic analyses, there are varying genetic and molecular studies that demonstrate rhizobia and NFs can induce defence signalling. For example, in *Medicago sativa* (alfalfa), *Sinorhizobium meliloti* inoculation triggers the production of reactive oxygen species (ROS), similar to what occurs in response to pathogen presence [52]. Moreover, a separate study on this same interaction demonstrated inoculation also induces a response characteristically similar to activation of plant immunity, including the accumulation of phenolic compounds and defence proteins [53]. In terms of NFs and their effects, please refer to the section below: “Nod factor signalling and suppression of MTI”.

## 3. The First Layer of Plant Innate Immunity

### 3.1. Rhizobia MAMPs and Evasion of MAMP-Triggered Immunity

The two-tiered plant innate immune system has been well studied in plant-pathogen interactions [54,55,56,57]. The first tier incorporates transmembrane pattern-recognition receptors (PRRs) at the plant cell surface that perceive microbe-associated molecular patterns (MAMPs, Figure 1). In the case of pathogen presence, these are termed pathogen-associated molecular patterns (PAMPs) [31,58]. MAMPs are conserved molecules not found in the host that are able to trigger a defence response known as MTI [29]. This is commonly characterised by the increased production of ROS, calcium influx, and immune signalling such as MAPK transduction to prime plant defences [37,59]. One well-known MAMP is chitin, a large structural polysaccharide consisting of N-acetylglucosamines. Chitin is found in a variety of plant pathogens and parasites. For example, fungal cell walls, arthropod exoskeletons, and a nematode pharynx all contain chitin as a key structural component [60,61,62]. As part of their immune system, plants have evolved to recognise the chitin substrate, leading to the priming of defence systems [63,64]. In response to the correct recognition of a pathogen, the plant host can secrete a wide range of defence-related proteins to block further invasion. These proteins can include both structural components such as cell wall maintenance, and antimicrobial or PR proteins such as phytoanticipins, proteinase inhibitors, and chitinases, which are hydrolytic enzymes that degrade chitin [65,66,67,68].

Nodulation is a highly specific process whereby plants have developed several mechanisms to restrict symbiosis with certain rhizobia genotypes [35]. There is accumulating evidence to suggest that MTI is one such mechanism that enables plants to discriminate beneficial rhizobia from those that are pathogenic. Mutualistic rhizobia contain many of the commonly found bacterial structures that act as MAMPs in plant-pathogenic interactions. These include flagellin and the surface polysaccharides lipopolysaccharides (LPSs), capsular polysaccharides (CPSs), and exopolysaccharides (EPSs) [27,69,70].

Bacterial flagellin acts as a potent PAMP in many reported plant-pathogen interactions, where it can be recognised by a flagellin receptor. One well characterised receptor is FLAGELLINSENSITIVE2 (FLS2), a leucine-rich repeat (LRR) receptor kinase (Table 1) [27]. FLS2 is conserved among numerous plant species, including *Arabidopsis thaliana*, *Solanum lycopersicum*, *Nicotiana benthamiana* (tobacco), and legumes such as soybean and *M. truncatula* [71,72,73,74,75,76]. The N-terminal region of the flagellin protein contains the flagellin 22 (flg22) epitope, which is responsible for FLS2-flagellin binding and is therefore critical for recognition [27]. Downstream of this interaction, plant defences such as MAPK signalling, ethylene accumulation, and other defences characteristic of an MTI response are activated [28].

Mutualistic rhizobia species also contain flagellin; however, sequence alignments revealed divergence in the N-terminal domain associated with flg22. This divergence appears to affect the recognition of rhizobia flagellins, making them inactive elicitors for *Arabidopsis* species [27] (Figure 1). Analysis of the *L. japonicus*-*M. loti* symbiosis found that purified flagellin of *M. loti* did not induce MAPK activation or ethylene accumulation in the host, suggesting the rhizobial protein does not act as a MAMP in this interaction [28]. This divergence would therefore enable rhizobial species to avoid recognition and MTI responses from the plant. It seems rhizobia have evolved to overcome MTI in this regard to facilitate invasion and nodule formation.

Other rhizobial MAMPs, such as EPSs, are crucial in establishing symbiosis and nodule formation. Early studies on EPS in nodulation found EPS-deficient mutants of *S. meliloti* to be defective in infection thread development and subsequent nodule formation on *M. truncatula* compared to wild-type rhizobia [95]. Inability to initiate infection threads with EPS-deficient rhizobia has also been reported for the *M. sativa*, *S. meliloti,* and pea- *Rhizobium leguminosarum* interactions [96,97]. It has been proposed that rhizobial EPSs can function to suppress host defences to facilitate invasion and nodule formation [95,98]. Alfalfa plants infected by an *S. meliloti* mutant deficient in EPS I, otherwise known as succinoglycan, produced pseudonodules with abnormally thicker cell wall structures, a phenotype often associated with resistance [99]. Moreover, these nodules contained a greater amount of phenolics compared to the wild-type strain. These compounds are characteristic of a general defence response and can be powerful inhibitors of pathogen growth [100]. Niehaus and Becker [69] proposed that these results were characteristic of a plant defence reaction and that this EPS is required for suppressing defences for normal nodule establishment. This is supported by a study by Aslam and colleagues [101], which investigated the EPSs of *S. meliloti* rhizobia for their ability to suppress host MTI. The authors found that recognition of MAMPs such as the bacterial flagellin peptide flg22 by the *A. thaliana* host triggers cytosolic calcium ion influx, which is required for the MTI response. *S. meliloti* EPS was shown to bind with calcium ions and consequently suppress downstream defences triggered by flg22 application [101]. It is interesting that EPS have this function given that rhizobia have a modified flg22 protein that does not trigger MTI [27]. This may indicate EPS may act as a general inhibitor of calcium influx instead of being specifically used for flg22-induced defence inhibition. Alternatively, calcium chelation by EPSs may have functioned to suppress MTI defences before rhizobial flagellin diverged and lost the flg22 epitope.

A co-inoculation-based analysis utilising nonsymbiotic bacteria demonstrated that EPS can also be important for nodule colonisation [102]. *L. japonicus* was co-infected with the nonsymbiotic bacteria *R. mesosinicum* KAW12 and the compatible *M. loti* R7A. In the presence of *M. loti* NF, KAW12 was able to colonise nodules [102]. It is interesting that KAW12 bacteria, which do not contain the gene clusters responsible for nodulation or nitrogen fixation, can act as endophytes when co-inoculated with compatible species. There is evidence to suggest that KAW12s EPS is important for its ability to colonise [102]. It is unknown whether KAW12 EPS functions in suppressing host defences. Further work to understand how KAW12 is able to evade an immune response in this interaction would improve our understanding of host-specificity in nodulation.

In *L. japonicus*, *M. loti* EPSs are perceived by exopolysaccharide receptor 3 (Epr3) and promote intracellular infection and nodule development [77,103] (Table 1; Figure 1). Epr3 is triggered by NF perception and enables the plant to monitor the various EPS structures secreted by different rhizobia strains. The compatibility of EPS with Epr3 leads to the promotion of bacterial entry and passage through the epidermis. However, recognition of incompatible EPS, such as truncated EPS, led to uninfected nodule primordia, but only in the presence of wild-type Epr3 and not in mutant lines [77]. This indicates that Epr3 can block infection with incompatible strains by an unknown mechanism. Together, these studies of rhizobial EPS demonstrate they play an important role in promoting infection during nodulation. Moreover, evidence of Epr3-EPS recognition suggests plants have evolved a secondary mechanism to monitor rhizobia compatibility for nodulation via MAMP recognition.

Less is known about the other surface polysaccharides, LPSs and CPSs, in the context of rhizobial infection and nodulation. For CPS, there is evidence that they can restore normal nitrogen-fixing nodules in the absence of compatible EPS [70]. Similarly, LPS have been shown to be important for rhizobial infection and primordia formation [37,104]. LPS are hypothesised to promote nodulation by suppressing defences due to their inhibitive effect on ROS [37,70]. For a variety of rhizobial MAMPs, including LPS, it appears they follow a common theme of evading MTI, suggesting that rhizobia have evolved to avoid recognition and subsequent defence responses from plant hosts.

Further studies focusing on the interplay between immunity and symbiosis utilised PAMPs alongside mutualistic rhizobia. For example, introduction of the *A. thaliana ELONGATION FACTOR-THERMO UNSTABLE RECEPTOR* (*AtEFR*), a PAMP receptor, into *M. truncatula* enabled perception of the EF-Tu PAMP epitope elf18 [105]. EFR recognition of elf18 confers resistance to the pathogenic bacterium *R. solanacearum*. The authors found that elf18-induced PTI could suppress *R. solanacearum* invasion while allowing nodulation to occur with *S. meliloti* [105]. The ability of rhizobia to infect roots in the presence of this defence reaction, which reduced pathogen infection, indicates that rhizobia are able to overcome the immune response.

However, this has not always been the case. In the *M. truncatula*-*S. medicae* interaction, nodulation is suppressed in the presence of the phytopathogen *R. solanacearum* [106]. Mutagenesis of the *R. solanacearum* T3SS revealed that this component plays a key role in nodulation suppression. It is possible that an effector secreted by the T3SS may be responsible for ETI and immune responses via its recognition by an intracellular receptor. Alternatively, the authors also suggest that suppression of PTI facilitated by the T3SS may induce defences leading to effector-triggered immune responses and suppression of nodulation [106].

### 3.2. Nod Factor Signalling and Suppression of MTI

While NF signalling is crucial for efficient symbiosis in nodulation, evidence suggests that it evolved from a primitive form of chitin recognition, and that the receptors have been repurposed for a mutualistic role [107,108,109]. Indeed, the utilisation of LCOs is not restricted to the legume-rhizobia symbiosis. LCOs are produced by a wide range of fungi species, including those that engage in mutualistic or pathogenic lifestyles. These structures are thought to be key components in regulating fungal growth and development [110]. This evidence indicates that LCOs are widely conserved as signalling molecules and can have diverse functions outside of nodulation. Structural observations comparing NF and chitin molecules support their proposed shared ancestry (Figure 2). Both proteins contain a backbone of N-acetylglucosamines [108]. However, there are several key differences that distinguish the two. NFs are generally smaller in chain length and can contain various modifications depending on the strain, including the addition of a fatty acid or 2-O-methylfucose [108].

Similar to chitin, NFs trigger a defence response, including the upregulation of defence genes such as chitinases, MAPKs, nucleotide binding site (NBS)-leucine-rich repeat receptors (LRRs) (NLRs), FLS2, and peroxidases [109] (Figure 1). For example, an uncharacterised soybean *NLR* gene exhibited transient upregulation induced by NF presence at early time points (12–48 hpi) [45] (Table 1). This timing corresponds with the reported transient upregulation of defence genes due to rhizobia infection in other datasets [42,44]. However, whether NFs or some other rhizobial factor is responsible for this defence upregulation reported in these latter datasets is currently unknown. Notably, while NF application results in transiently induced plant defences, the expression level of these defences was considerably lower than those induced by chitin oligosaccharides [109].

NFs also induced ROS production in various interactions, including *M. truncatula*, *S. meliloti*, *M. sativa*-*S. meliloti,* and *Phaseolus vulgaris* (common bean)-*Rhizobium etli* [52,78,111]. For example, compatible NFs of *S. meliloti* were found to rapidly induce ROS production in the epidermal root cells of its *M. truncatula* host [111]. This localised production of ROS was not observed in *S. meliloti* mutants, which produce NFs lacking a sulphate moiety required for rhizobial infection and nodule formation [111]. ROS production in this interaction has been shown to promote symbiosis by facilitating infection thread formation and progression [112]. This appears to also be the case for the common bean-*R. tropici* interaction. Respiratory burst oxidase homologs B (RbohB) is classified as a NADPH oxidase, enzymes that are known for their role in ROS production. RbohB was found to promote rhizobial colonisation, nodulation, and nitrogen fixation [78,79] (Table 1). NFs also appear to regulate pathogen-induced ROS [111]. In the *M. truncatula–Aphanomyces euteiches* phytopathogenic interaction, ROS production was impeded by NF signalling [113]. Therefore, while they are considered antimicrobial compounds in plant-pathogen interactions, in nodulation they are thought to be an essential component for facilitating symbiosis [114,115]. Moreover, unlike other defences, NF induction of ROS may therefore be a mechanism to facilitate symbiosis instead of MTI induction.

Aside from NFs, the NFRs from *M. truncatula* and *L. japonicus* have been shown to induce defence reactions in tobacco, such as cell death [116,117]. Further studies in *M. truncatula* found that ectopic expression of *MtNFP* in nodule tissue led to a higher density of uninfected cells, while infected cells demonstrated signs of premature cell death [118].

While NFs can induce defence gene upregulation via NFRs, they can also actively suppress MTI to facilitate the symbiosis via a mechanism independent of NFR perception (Figure 1) [109,119]. In soybean mutants unable to express NFR1/5, application of *B. diazoefficiens* USDA110 NFs was able to significantly reduce the production of ROS and MAPK phosphorylation triggered by flg22 [119]. Moreover, an MTI-suppressive effect by NFs was also demonstrated in the nonlegume *A. thaliana* [119]. These results prompted questions about what receptors or receptor complexes are perceiving NFs aside from NFRs to facilitate the suppressive effect. A lysin motif receptor-like kinase (LYK) in *Arabidopsis* termed LYK3, which is related to NFR1/5 in soybean, was discovered to be required for NF-induced suppression of flg22-induced ROS production (Figure 1) [119,120]. Therefore, while NFs act to induce MTI via NFR receptors in soybean, in *Arabidopsis* they appear to suppress these defences with a similar receptor kinase, LYK3.

Aside from NF molecules, Liang et al. [119] tested a variation of chitin oligomers for a suppressive effect on flg22-induced ROS signalling. Shorter chitooligosaccharides of 4–5 dp were able to reduce ROS signalling, yet not as effectively as NFs [119]. However, longer chitin molecules (6–8 degrees of polymerisation (dp)), which are known to induce MAMP signalling did not suppress ROS production. These results suggest that a shorter chitin oligomer length as well as NF modifications appear to enhance the suppressive effect. Liang et al. [108] hypothesised that fungal pathogens could evade MTI by hydrolysing their MAMP-inducing chitin molecules, creating short-chain chitin that suppresses defences.

With current evidence to suggest chitin and NF recognition share common ancestry, it is possible that these NF attributes, including shorter chain length, may be the result of an evolutionary adaptation, whereby rhizobial NFs were derived from a more primitive chitin form to lower plant defences and enable infection [107,109]. This is supported by the analysis of the NF and chitin receptors *LjNFR1* and *Chitin Elicitor Receptor Kinase 1* (*AtCERK1*), respectively. These two receptor proteins are structurally similar, with high sequence similarity in their intracellular kinase domains, and are believed to have evolved from a common ancestor [109,121]. NF-induced MTI is dependent on *LjNFR1,* and Nakagawa et al. [109] propose that this response is only apparent due to their close ancestry to chitin and *AtCERK1,* respectively, and that NF-induced MTI is regulated by other components of NF signalling. Therefore, while NF-induced MTI is observable, the authors propose it is an artefact of an older signalling mechanism that is no longer required and is suppressed by NF signalling. Transcriptomic analyses of nodulation have observed a transient defence response after rhizobial inoculation, which supports this view [42,44,45]. The parallels between NF and chitin signalling are also apparent in the comparison of *M. truncatula* NFRs *MtNFP* and *MtLYK* with *AtCERK1*. Expression of *AtCERK1* produced a similar effect to the co-expression of the NFRs *MtNFP* and *MtLYK* in *N. benthamiana,* where both treatments induced cell death [116].

Similar to chitin molecules, NFs can be degraded by chitinases (Figure 2). These enzymes are well-known for their role in plant defence, where they function to degrade the chitin backbone of fungal pathogens via hydrolysis of their glycosidic bonds. However, they appear to function positively in nodulation [86,122]. The *M. truncatula* NOD FACTOR HYDROLASE1 (NFH1) enzyme has been shown to preferentially hydrolyse NFs instead of chitin [80,81]. Moreover, mutant *nfh1* analysis demonstrated delayed root hair infection, for which Cai et al. [81] suggest NFH1 may have a role in regulating NF to avoid excessive levels accumulating, which may negatively impact infection. Several other examples of NF-hydrolysing chitinases exist (Table 1), including *S. rostrata chitinase 13* (*SrChi13*), which is induced upon infection in stem nodules [82]. However, further work is needed to address its biological importance and whether it plays a similar role to *MtNFH1*. Taken together, evidence of chitinases targeting NFs further supports the evolution of NFs and chitin oligosaccharides from a common ancestor. Furthermore, it demonstrates how the plant immune system plays an important role in rhizobia infection and nodulation.

Interestingly, plant immune responses induced upon rhizobial inoculation can also be suppressed in a NF-independent manner. The symbiosis receptor-like kinase (SymRK) of *L. japonicus* is required for the suppression of the transient MTI response upon rhizobial treatment [123,124]. SymRK is proposed to associate with BRASSINOSTEROID-INSENSITIVE1-Associated receptor Kinase 1 (*LjBAK1*), a positive regulator of plant immunity, and inhibit its kinase activity. *LjBAK1* mutants demonstrated increased numbers of infection pockets and infection threads upon *M. loti* treatment, indicating that its loss of function promotes rhizobial infection [124]. More recently, a novel *L. japonicus* protein NONRACE-SPECIFIC DISEASE RESISANCE1/HARPIN-INDUCED1-LIKE13 (NHL13), was found to associate with SymRK. This protein is similar to *Arabidopsis* NHL, which can activate immune responses [125]. NHL mutants of *L. japonicus* exhibited weaker infection by *M. loti* as well as higher expression of defence-related genes compared to the WT, suggesting they promote infection and reduce plant immunity [125]. Taken together, these investigations of SYMRK, BAK1, and NHL13 provide evidence of NF-independent suppression of plant immunity and reveal, in part, how plant immunity and symbiosis overlap in nodulation.

## 4. The Second Layer of Plant Innate Immunity

### 4.1. Rhizobia Have Adopted Pathogenic Machinery to Suppress Plant Immunity and Prime the Plant for Colonisation

In order to suppress host defences triggered by PAMPs and facilitate further invasion, pathogenic microbes have evolved to produce and secrete virulence proteins termed effectors into the host cell. These effectors are often translocated into the plant cytoplasm via bacterial appendages such as the type III secretion system (T3SS), the type IV secretion system (T4SS), and the type VI secretion system (T6SS) [126,127,128]. Certain effectors have the ability to hijack host cellular processes and machinery to inhibit PTI signalling and promote infection once again [129]. Indeed, inactivation of *R. solanacearum*’s T3SS was shown to be one of the two crucial mutations to enable this pathogenic rhizobia species to successfully infect and nodulate the legume host, *Mimosa pudica* [130].

Analysis of secretion systems within different rhizobia species has revealed that the T3SS, the T4SS, and the T6SS are all present to some extent within this group of nitrogen-fixing microsymbionts [126]. Further detail regarding the role of secretion systems and the rhizobia that possess them has been reviewed previously [131]. Interestingly, while not all rhizobia contain these functional secretion systems and effectors, the certain species that do can utilise them to suppress defences and promote infection, similar to plant-pathogen infection [92,126,132,133] (Figure 1). Upon flavonoid perception, NodD induces TtsI, which initiates T3SS transcription [126]. Rhizobia effectors within the nodulation symbiosis are termed nodulation outer proteins (Nops), many of which are structurally similar to effectors secreted by pathogenic bacteria [36]. Most of these proteins are expressed during early infection and in mature nodules and function to promote nodulation [36,134]. For example, the NopL protein from broad-host-range *Sinorhizobium* strain NGR234 has a positive role in nodulation. Mutant NopL rhizobia induced fewer nodules on the legume host *Flemingia congesta*, while overexpression of the NopL protein suppressed host immunity by impairing MAPK signalling in yeast and transgenic tobacco plants [135,136]. Similarly, NopM, an E3 ubiquitin ligase secreted by the *Sinorhizobium* strain NGR234, suppressed plant defences and promoted nodulation in tobacco [137]. Specifically, expression of NopM in tobacco was found to reduce the level of ROS produced in response to the bacterial flagellin peptide flg22 [137]. The authors hypothesised that suppressing ROS production by NopM during infection enables further nodule initiation.

### 4.2. The T3SS Can Induce Nodulation Independently of NFs

While the rhizobial T3SS and its secreted effectors are important for interfering with and suppressing immune signalling, additional evidence suggests they also manipulate nodulation signalling by a mechanism independent of NFRs. In soybean cv. Enrei, the *Bradyrhizobium* strain USDA61 induced nodulation in the absence of a functional plant NFR [132]. In this study, *nodC^-^* mutants of USDA61, which are unable to synthesise NFs, induced nodulation in WT and *nfr1* mutant plants. However, T3SS-deficient rhizobia were not able to induce normal nitrogen-fixing nodules in *nfr1* mutants [132]. Together, these results suggest normal NF signalling is not required for effective nodulation in this interaction when the T3SS is functional.

Interestingly, in *nfr1* mutants inoculated with WT rhizobia, there was an observed absence of root hair curling and infection threads [132]. Okazaki et al. [132] propose a model for this symbiosis whereby rhizobia are able to bypass normal early nodulation signalling events when NFs are absent and instead colonise soybean plants via primitive forms of rhizobial entry such as crack-root or intracellular entry. Okazaki et al. [138] further suggested that it might have been an early approach of colonisation during the evolution of legume-rhizobia symbiosis, which has been retained yet superseded by the more efficient NF signalling.

Further investigations following this study focused on identifying T3SS effectors involved in this NF-independent pathway. Teulet and colleagues [139] identified several effectors within the *Bradyrhizobium* sp. ORS3257-*Aeschynomene indica* interaction. The proteins promoted NF-independent nodulation with distinct roles, some suppressing defence responses, others involved in nodule formation. One well-characterised example is the Bel2-5 effector of *B. elkanii* USDA61, which enables the strain to induce nodulation in *nfr1* mutants of soybean [92]. Interestingly, this protein closely resembles the XopD effector from the phytopathogen *Xanthamonas capestris*. It is hypothesised that Bel2-5 has a similar role to XopD in downregulating host defences via the suppression of ethylene biosynthesis, but more work is needed to understand its functionality [92]. Together, these findings further emphasise the importance of effectors and the T3SS in nodulation and suggest certain rhizobia have adopted this NF-independent pathogenic system to deliver effectors into the host cell and hijack nodulation signalling.

Alternatively, there exist certain bacteria that lack both T3SS and nod genes but still nodulate plants of the *Aeschynomene* genus [138,140]. This indicates a third pathway independent of NF and the T3SS must exist; however, further work is needed to uncover the underlying mechanisms employed here.

### 4.3. Legumes Employ R Genes to Recognise Rhizobia Effectors and Block Invasion

Plant-pathogen co-evolution has equipped the plant immune system with a second layer of defence to combat effector-secreting pathogens, culminating in effector-triggered immunity (ETI) [127]. This response relies on gene-for-gene specificity, whereby effectors are recognised by their co-evolved plant intracellular receptors termed R proteins [141]. These proteins usually have high specificity for their cognate effectors compared to the non-specific host resistance of PTI [36]. Activation of ETI often results in a hypersensitive response (HR), which is characterised by localised plant cell death at the infection site to avoid further spread of the pathogen [54].

Evolutionary pressures drive the modifications in the suite of effectors secreted by pathogens to avoid ETI, once again overcoming host defences. These changes include shedding the effector gene triggering ETI, or mutations in this effector that evade recognition by its cognate *R* gene. Additionally, the pathogen can acquire new virulence proteins via evolutionary mechanisms such as horizontal gene transfer, which would not yet have a co-evolved *R* gene [36,54,127]. However, natural selection favours the creation of new *R* genes to recognise new virulence proteins, leading to ETI once again [54]. Ultimately, these antagonist interactions result in ongoing cycles of ETI and effector-triggered susceptibility (ETS), leading to the co-evolution of effector-*R* gene specificities whereby both species are continuously diversifying and adapting to survive [56].

NLRs constitute a large family of *R* genes and are well-known key players in effector recognition. The family is subdivided by distinct differences in domain organisation at the N-terminus. The presence of either a Toll/interleukin-1 receptor (TIR) or coiled-coil (CC) domain at the N-terminal end classifies an NBS-LRR protein as either an TIR-NBS-LRR (TNL) or CC-NBS-LRR (CNL), respectively [142]. Interestingly, while CNLs are present in both monocotyledons and dicotyledons, TNLs have so far only been identified in dicotyledon species [142,143,144]. While structurally distinct, these protein subfamilies can function synergistically in plant immunity. Members within the TNL and CNL subgroups are known to directly or indirectly perceive effectors and are thus classified as sensor NLRs (sNLRs) [145]. Moreover, recent investigations have revealed that the TIR domains of certain NLRs act as NADase enzymes to promote cell death. For example, RESPONSE TO HOPBA1 PROTEIN (RBA1), RECOGNITION OF *PERONOSPORA PARASITICA* 1 (RPP1), RESISTANCE TO *PSEUDOMONAS SYRINGAE* 4 (RPS4), and RESISTANCE TO *UNCINULA NECATOR* PROTEIN (RUN1) have all been shown to confer NAD+ degradation [146,147].

Additionally, certain CNLs can function downstream of sensors as “helper NLRs” (hNLRs) to transduce immune signalling from the activated sensor NLRs to yet unknown components [148]. hNLRS are classified into three families: the ACTIVATED DISEASE RESISTANCE 1 (ADR1) family [149], the N REQUIREMENT GENE 1 (NRG1) family [150], and the NB-LRR protein required for HR-associated cell death (NRC) family [151]. *ADR1* and *NRG1* gene members contain CC domains closely related to the RESISTANCE OF THE *POWDERY MILDEW 8* (*RPW8*) gene in *A. thaliana* and are thus categorised as RPW8-NBS-LRRs (RNLs) [152]. hNLR members have been identified in a wide spectrum of plant-pathogen interactions and are critical components for facilitating ETI signalling downstream of several TNLs and CNLs, including RESISTANCE TO *PSEUDOMONAS SYRINGAE* PROTEIN 2 (RPS2), RPS4/ RESISTANCE TO *RALSTONIA SOLANACEARUM* 1 (RRS1), Recognition of XopQ 1 (Roq1), and more [148]. While both CNLs and TNLs have been shown to induce cell death, the complete set of downstream components still requires further elucidation.

*R* genes play a similar role in nodulation, where their encoded receptors are able to recognise secreted Nops [134] (Figure 1). Therefore, while Nops can facilitate rhizobial invasion in some hosts, their indirect recognition by plants containing certain nodulation-specific R proteins can result in ETI, preventing successful infection and symbiosis [87,93]. In this way, *R* genes play a crucial role in determining host-specificity by limiting the number of strains that can successfully infect the host.

This *R* gene-mediated defence may explain the variation in the effects Nops can produce in nodulation. For example, the *incompatible nodulation B (innB*) gene of *B. elkanii* strain USDA61 promotes symbiosis in *Vigna radiata*, yet it is the cause of incompatibility with many other *Vigna* species [153]. Although no underlying mechanism for the inhibition has been identified, it would be interesting to investigate potential *Vigna* sp. *R* genes for a role in this interaction. While there have been several functionally characterised *R* genes in nodulation [49,89,93], all so far have been identified and studied in soybean. Therefore, our understanding of *R* genes in other legume species and how they are utilised in nodulation is limited to this one species.

*R* genes were first discovered to play a role in nodulation by Yang et al. [49]. The allelic *Rhizobium japonicum 2/Rhizobium fast-growing 1 (Rj2/Rfg1) TIR-NBS-LRR (TNL)* genes were shown to determine host-specificity of soybean by restricting nodulation with strains *B. diazoefficiens* USDA122 and *S. fredii* USDA257, respectively [49] (Table 1). Rj2 recognises the effector NopP from an incompatible strain, *B. diazoefficiens* USDA122. Perception upregulates defence responses, leading to ETI to inhibit further invasion of the microbe [49,87]. Likewise, cultivars carrying the *Rfg1* allelic variant restrict nodulation with certain *S. fredii* strains, such as USDA257 [88]. While Rfg1 appears to act similarly to Rj2, less has been shown about its functionality in soybean-*S. fredii* interactions.

Additional studies discovered *Rhizobium japonicum 4* (*Rj4)*, a dominant gene in soybean encoding a thaumatin-like protein [89]. Rj4 functions to restrict nodulation with certain highly competitive strains of *B. diazoefficiens* and *B. elkanii* [89,91] (Table 1). Transcriptomic analyses revealed that the presence of the *B. elkanii* T3SS upregulates defence-related genes characteristic of the ETI response in soybean cv. BARC-2, which contains the dominant *Rj4* allele [154]. Further studies found that the secreted *B. elkanii* effector Bel2-5 was responsible for the incompatibility with plants carrying the *Rj4* allele [90,133]. It seems that while the Bel2-5 effector can facilitate NF-independent nodulation in some *nfr1* mutant plants, those that contain the *Rj4* allele are able to recognise the effector and promote defence signalling, which inhibits further infection and nodule formation [133].

*Nodule Number Locus 1 (GmNNL1)* was more recently identified in soybeans as a novel *TNL* gene which functions to bind the effector NopP from *B. diazoeffeiciens* USDA110 and activate defence responses leading to the inhibition of nodulation via root hair infection [93] (Table 1). Notably, when this occurs, rhizobia can still infect via crack-root entry, which results in a significantly lower nodule number than root hair infection. Interestingly, certain *NNL1* haplotypes encode a truncated version of the gene that is unable to recognise NopP [93]. This avoids the ETI response, enabling soybeans to undergo root hair curling and infection thread formation. Further evidence from this study suggests soybean has evolved to contain an insertion by *GmSINE1,* which leads to this truncated version. Plant haplotypes containing the insertions are more favourable because they can bypass ETI and utilise the more sophisticated and efficient method of entry via root-hair curling as opposed to crack-root entry [93].

It is interesting that both nodulation-related NLRs, NNL1 and Rj2, recognise the effector NopP in USDA110 and USDA122, respectively [87,93]. It has been shown that small variations in the protein sequence of NopP can account for Rj2 recognition and subsequent incompatibility [87]. Therefore, variances in NopP between USDA110 and USDA122 may account for recognition by different proteins. Moreover, NNL1 was shown to recognise NopP not by the LRR or post-LRR domains, which are reported for Rj2 [87], but by the TIR domain. It appears these TNLs have very different recognition and signalling mechanisms for recognising the same Nop protein, which broadens our current understanding of TNL activity in nodulation. Whether the TIR domains of Rj2 and NNL2 function to transduce immune signalling via NADase activity is not known. Incompatible interactions were found to induce HR with either protein. However, NNL1 cannot induce HR with the TIR domain alone such as other reported TNLs, as the LRR domain is required for this functionality [93,155]. Further investigations focused on the downstream components would enhance our current understanding of the functionality of each protein’s domains.

### 4.4. Calcium Signalling Regulates the Legume-Rhizobia Symbiosis, Could NLRs Be Involved?

Plant NLRs have well-characterised roles in effector perception and immune signalling for inducing HR. However, recent structural insights carried out on a wide range of *NLR* genes demonstrate they can form novel oligomeric structures called resistosomes with unique roles in calcium signalling, redefining NLR functionality in plant immunity. For example, upon effector-induced activation, the *HOPZ-ACTIVATED RESISTANCE 1 (ZAR1)* sensor *CNL* gene product of *Arabidopsis* can form a pentameric resistosome, which facilitates cell death [156]. Further investigations found that the N-terminal helices of this complex form a funnel-shaped structure that functions as a calcium-permeable cation channel [157,158]. Plant-pathogen interactions are often accompanied by calcium signatures such as the rapid influx of calcium ions (Ca^2+^) in the cytosol upon pathogen presence, which is often considered a hallmark of infection [159,160]. Moreover, in NLR signalling pathways, Ca^2+^ is often observed downstream of NLR activation and has been shown to be critical for NLR signalling and inducing HR [161,162]. Bi et al. [158] demonstrated that ZAR1 activation triggers calcium influx in the cytosol, the production of ROS, and, lastly, cell death. Together, this evidence suggests that NLRs can play a novel role in inducing cell death by acting as calcium-permeable channels to facilitate cytosolic calcium levels upon infection.

In addition to this sensor CNL, hNLRs and sensor TNLs have also been implicated in calcium signalling in plant-pathogen interactions. hNLRs have been shown to induce cytosolic calcium influx and subsequently trigger cell death in *Arabidopsis* [163]. Moreover, given hNLRs have been shown to be a requirement for all currently tested TNLs. TNLs have been hypothesised to be an upstream player of hNLR activation here. TNLs may also contribute to calcium signalling by acting as NADase enzymes. NAD+ degradation produces several products, including ADP ribose (ADPR), a variant of cyclic ADPR (v-cadpr), and nicotinamide. Wan et al. [157] propose that NADase activity may be important for pathogen-induced calcium signalling, as the products, ADPR and cyclic ADPR, are known to trigger cytosolic calcium influx [164,165]. Together, these recent findings broaden our understanding of NLRs with novel functionalities such as calcium channels and NAD^+^ cleaving enzymes. Furthermore, they suggest that these diverse gene subfamilies are working synergistically to regulate calcium signalling and, therefore, cell death in response to pathogen presence. While these NLRs function in plant immunity, their roles in calcium signalling make them interesting targets for studying calcium signalling in nodulation.

In legume nodulation, NFs trigger a myriad of responses, including two well-known calcium signalling events: cytosolic Ca^2+^ influx and oscillations in cytosolic Ca^2+^ (calcium spiking) [166]. These responses have been reported in many of the model legume species, including common bean, pea, *M. truncatula,* and *L. japonicus* [167,168,169,170]. While they are both related to nodulation, it is believed they are functionally separable and belong to distinct signalling pathways. Cytosolic Ca^2+^ influx occurs first in a rapid burst at the root hair tip within the first 5 minutes of NF application in *L. japonicus* [167,170]. In contrast, Ca^2+^ spiking occurs in the nuclear region of the root hair cell and occurs 10 min after NF application [166,167,168,169,170,171]. Importantly, spiking can be induced at much lower NF concentrations (1 nM), while Ca^2+^ influx has been observed only at 10 nM [166,172]. Miwa et al. [172] propose that, in natural conditions, Ca^2+^ spiking may occur first when rhizobia begin to infect, as bacterial concentration increases during infection. The authors propose the NF concentration increases to a high enough level to trigger a Ca^2+^ influx. This is supported by evidence that Ca^2+^ influx can be induced after Ca^2+^ spiking [166].

Ca^2+^ spiking is thought to be essential for nodulation by facilitating NF signalling and nodule organogenesis [11]. Conversely, Ca^2+^ influx is thought to have a role in infection rather than nodule formation. The extent of Ca^2+^ influx is greatly reduced in *M. truncatula* when inoculated with *S. meliloti nodl* mutants that exhibit reduced infection [173]. Ca^2+^ spiking was not influenced by this same mutant, providing further genetic evidence that these two signatures reside within distinct pathways. Moreover, ethylene, which inhibits infection thread development, was also shown to reduce Ca^2+^ influx [173,174].

That Ca^2+^ influx is important for rhizobia infection seems somewhat surprising given the similar Ca^2+^ signature in plant-pathogen interactions, which activates downstream immunity signalling to induce HR and therefore block further invasion. However, Ca^2+^ signatures across plant-microbe interactions can vary in amplitude, location, and duration, and these factors influence the specificity of the Ca^2+^ response [175]. This variance is important in enabling the plant to induce immunity or symbiosis signalling [176]. Therefore, while nodulation may employ a similar signature in terms of timing and duration, there still exists some variance that allows it to trigger distinct responses.

Given the number of similarities between the two infection pathways outlined throughout this review, it is possible that nodulation may recruit similar NLR players in inducing rhizobial infection as it has done for plant-pathogen interactions by utilising ZAR1 and hNLRs [148,158] (Figure 1). In the *S. rostrata-A. caulinodans* interaction, intracellular colonisation was associated with localised plant cell death and the production of ROS [177]. The latter was found to promote nodulation by mediating NF signalling. Both cell death and ROS are associated with calcium signalling and NLR activation in plant-pathogen interactions, indicating a similar signalling process may occur in nodulation [160]. For example, the active ZAR1 calcium-permeable channel was found to be required for ROS signalling and cell death [158].

The observation that NLRs can facilitate Ca^2+^ influx, suggests that they are important candidates for investigating this activity in nodulation, especially considering Ca^2+^ influx is essential for rhizobia infection. Whether their roles are related to blocking invasion of incompatible species by activating HR via Ca^2+^ signalling or alternatively facilitating Ca^2+^ influx and/or Ca^2+^ spiking to increase infection and nodulation is an important question to address.

## 5. Conclusions and Future Directions

While nodulation is a mutualistic interaction, the plant’s innate immune system is triggered during early infection, similar to what occurs in a plant-pathogen interaction [42,43,44,45] (Figure 1). Yet the function and relative importance of this defence response in nodulation is not well understood. There is evidence to suggest that mutualistic rhizobia have co-evolved with plants to facilitate nodulation by employing the same mechanisms used by pathogenic rhizobia to overcome plant immunity, while an immune response is inadvertently triggered by NF perception [36,109]. Indeed, it has been proposed that this triggered defence response is a relic of pathogenic signalling from which the legume-rhizobia symbiosis may have evolved and that it needs to be suppressed for nodulation to succeed [36,42]. In this case, results from transcriptomic analyses of compatible symbionts that did not induce a defence response [50,51] may highlight a more efficient infection process where defences do not need to be induced. Still, it can be argued that the plant immune system plays a critical functional role in nodulation. Proteins such as chitinases and EPS receptors play functional roles and are essential for infection [77,80]. NFs also induce the production of ROS, and while ROS is detrimental in plant-pathogen interactions, it is suggested to have a positive effect in the legume-rhizobia symbiosis [78,177]. Establishing differences in the plant’s immune response following infection with certain rhizobia that invade via crack entry, as opposed to root-hair invasion, could further help delineate the role of key molecular factors acting in these processes [178]. Moreover, determining how the plant host can identify and sanction rhizobia based on their nitrogen fixation capacity is also of great interest for better understanding [179].

Considering NFs trigger the upregulation of certain NLRs, it would be interesting to examine if any of these NLR proteins were important for host-specificity or nodulation, such as NNL1 or Rj2. If so, it would suggest another function for the initial defence response in the nodulation process. Similarly, with NLRs implicated in Ca^2+^ influx that is induced by NFs, structural analyses to establish whether nodulation-related NLRs play similar roles in Ca^2+^ signalling and NAD^+^ degradation would provide further insight into the plant immune system’s involvement in nodulation. Moreover, with only a few NLRs so far identified as playing a role in soybean nodulation, further work to identify and characterise additional NLRs in other species, and those having different domain organisations, would broaden our understanding of defence signalling pathways in nodulation. CNLs and helper NLRs work synergistically with TNLs, and a similar interaction may be important for host-specificity with nodulation TNLs.

With limited knowledge of the plant immunity proteins that influence nodulation, it is clear that this area requires further attention. Future investigations of the factors involved in determining genetic compatibility and successful infection, both within the plant immune system and rhizobia infection mechanism, could benefit agricultural management practises and support the selection of superior rhizobia strains and crop varieties that enhance effective nodulation in the field.

## Figures and Tables

**Figure 2 ijms-24-02800-f002:**
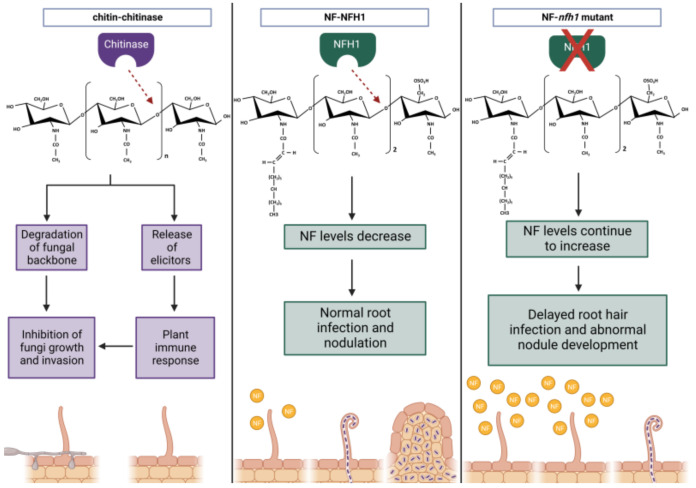
A comparison of the proposed roles of chitinases targeting the glycosidic linkages in either chitin or NFs for degradation. In response to a phytopathogenic fungi, the plant releases chitinases, which function to degrade the chitin backbone and/or release elicitors to further upregulate defences. In response to NFs, chitinase NOD FACTOR HYDROLASE1 (NFH1) degrades NFs to regulate NF levels and optimise root hair infection. In the absence of NFH1, NF levels are unregulated, which results in delayed root-hair infection and abnormal nodule development. NF—Nod Factor and NFH1—NOD FACTOR HYDROLASE1. Created with BioRender.com.

**Table 1 ijms-24-02800-t001:** Legume resistance proteins currently identified or proposed to function within MTI or ETI, demonstrating a known effect on or role in nodulation.

Gene Name	Host Species	Bacteria Species	Protein Family	Putative or Known Function	MTI vs. ETI	References
*FLAGELLIN SENSING2 (FLS2)*	*L. japonicus*	*M. loti*	Receptor kinase	Perceives flagellin containing the active epitope flg22, and induced defence responses leading to inhibition of rhizobial infection and delayed nodule formation.	MTI	[28]
*Exopolysaccharide receptor 3 (Epr3)*	*L. japonicus*	*M. loti*	LysM serine/threonine receptor kinase	Perceives *M. loti* EPS and determines compatibility for symbiosis.	MTI	[77]
*Respiratory burst oxidative homolog B (RbohB)*	*P. vulgaris*	*Rhizobium tropici*	NADPH oxidases	Facilitates ROS production and positively regulates rhizobia colonisation and nodulation.	MTI	[78,79]
*NOD FACTOR HYDROLASE1 (NFH1)*	*M. truncatula*	*S. meliloti*	Chitinase	Regulates NF-levels for normal root hair infection by hydrolysing NFs.	MTI	[80,81]
*Chitinase 13 (Chi13)*	*S. rostrata*	*Azorhizobium caulinodans*	Chitinase	Hydrolyses NFs with NF-specific gene expression. Putative role in nodule ontogeny.	MTI	[82,83]
*Chitinase 24 (Chi24)*	*S. rostrata*	*A. caulinodans*	Chitinase	NF-specific gene expression with a putative role in NF binding.	MTI	[84]
*CHITINASE 24 (CHIT24)*	*M. sativa*	*S. meliloti*	Chitinase	Hydrolyses NFs and chitin with unknown effects on nodulation.	MTI	[85]
*CHITINASE 36 (CHIT36)*	*M. sativa*	*S. meliloti*	Chitinase	Hydrolyses NFs with unknown role in nodulation.	MTI	[85]
*Chitinase 30 (Chi30)*	*M. sativa*, *Vicia sativa*, *P. vulgaris*	*S. meliloti*	Chitinase	Hydrolyses NFs with unknown role in nodulation.	MTI	[13]
*CHITINASE 5 (CHIT5)*	*L. japonicus*	*M. loti*	Chitinase	Hydrolyses NFs to facilitate primordia infection.	MTI	[86]
*TIR-NBS-LRR (uncharacterised)*	*G. max*	*Bradyrhizobium diazoeffeciens*	*R* gene; TNL	Upregulated in response to strain CB1809 in the zone of nodulation. Expression is NF-specific.	ETI	[45]
*Rhizobium japonicum 2 (Rj2)*	*G. max*	*B. diazoeffeciens*	*R* gene; *TNL*	Restricts nodulation with USDA122 by recognising effector NopP.	ETI	[49,87]
*Rhizobium fast-growing 1 (Rfg1)*	*G. max*	*S. fredii*	*R* gene; *TNL*	Restricts nodulation with USDA257 and USDA193 by unknown mechanisms.	ETI	[49,88]
*Rhizobium japonicum 4 (Rj4)*	*G. max*	*B. elkanii*	*R* gene; *TNL*	Restricts nodulation with USDA61 by recognising effector Bel2-5.	ETI	[89,90,91,92]
*Nodule Number Locus 1 (NNL1)*	*G. max*	*B. diazoeffeciens*	*R* gene; *TNL*	Recognises effector NopP and triggers defence responses to inhibit nodulation via root hair infection.	ETI	[93]
*MAP kinase kinase 5 (MP2K5- MAP kinase 3/6 (MPK3/6)*	*M. truncatula*	*S. meliloti*	MAP kinase (MAPK) and MAP kinase kinase (MAP2K)	This MAPK signalling module functions to negatively regulate nodulation formation.	MTI	[94]

## Data Availability

Not applicable.

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
