# Peer review of "Legumes Regulate Symbiosis with Rhizobia via Their Innate Immune System"

_ijms, 2023, doi:10.3390/ijms24032800_

Round 1
Reviewer 1 Report
Dear Authors, I fully recommend your manuscript to be published at present form.
Author Response
We thank the reviewer for their recommendation of our manuscript. Please see the attached manuscript for resubmission.

Reviewer 2 Report
This is an area that has recently reviewed extensively so I was somewhat skeptical of the need for another such review. However, progresses in our understanding of plant innate immunity make such a proposition timely and I commend the authors for bringing in some new ideas and analysis. The review is very well written. It assembles and organizes available information about the role of plant innate immunity during rhizobia-legume interactions. My main concern is that some important recent symbiosis-related papers are not discussed or even mentioned while a numbers of references are rather “old” –mainly reviews papers- standing for general description of what is known on plant innate immunity -could be removed as there are redundancies. In addition, one may miss the description of the control of plant immunity after the infection process but this was likely a choice of the authors to focus on very early stage of the interaction.
My remarks and suggestions:
Here are some recent papers that are tatally in the scope and that are important ones:
-1- Yamazaki et al MPMI 2022 (doi: 10.1094/MPMI-11-21-0263-R.)
-2- Feng et al Mol Plant 2021 (doi: 10.1016/j.molp.2021.07.016)
-3- About the role of MTI in symbiosis, the authors might consider to mentioned the work reported by Pfeilmeier et al Plant biotechnol 2019 (doi: 10.1111/pbi.12999 heterologous expression of EFR in Medicago, the rhizobia display the elf18 peptide in the EF-TU protein…).
-4- Line 175-178: I am concerned that the formulation here might be misleading. To my knowledge, no MAMP has been isolated from rhizobia as effective to trigger defense reactions on its host plant (Berrabah et al MPMI 2019 doi: 10.1094/MPMI-07-18-0205-FI).
-5- About nod factor/LCO, the authors might consider to include in their discussion the fact that LCO are not only produced by mutuasltic fungus (Tomas Allen Rush et al Nature communications 2020 doi: 10.1038/s41467-020-17615-5).
-6- Two relatively recent papers deal with innate immunity and rhizobia-legume interactions and are particular in the sense that they include works with pathogens: the NF-focused Rey et al paper published in New phytologist (2019 doi: 10.1111/nph.15574.)
-7- the pathogen-mediated restriction of nodulation-focused Benezech et al MPMI paper (2021 doi: 10.1094/MPMI-11-20-0319-SC.). The authors might consider to mention those works.
-8- About the title, Legume regulate symbiosis with rhizobia via their innate immune system but not only (NF perception, AON…), the authors might to make it more baanced.
-9- line 46 ”These organs, induced by the microsymbiont (mainly bacteria broadly called rhizobia)”. I don’t understand why the word “mainly” is employed her; are there other bacteria that would not be rhizobia that trigger nodules?
-10- Line 48/49: plant hosts do not provide only carbohydrates to the rhizobia, plants provide every macro- and micro-nutrients required for the metabolism of rhizobia.
-11- Emphasis is putted on the roles of T3SS and NLRs and the authors might consider making clear that not all rhizobia display such secretion systems. For instance, to my knowledge, none of the Medicago symbionts has T3SS –except the 1022 strain-. In addition, it has been shown that T4SS can also translocate effectors in plant cells and that, in some cases it contributes to nodulation (see the work from M. Sadowski lab).
-12- The importance of the T3SS system in restricting nodulation was pretty well demonstrated by the group of Catherine Masson years ago and the authors might consider mentioning this work –Marchetti et al 2010 PloS biology paper-.
-13- The authors might consider mentioning that legume plants exert an active control on their immune status after rhizobia internalization in the symbiotic cells (see the literature about DNF2/NAD1/RSD/SymCRK).
-14- At the end of the manuscript, the authors address the question of a potential positive contribution of NLR in infection –they do focus on their action on calcium fluxes-. With this respect, they might consider to mention that during the interaction of Aeschynomene and Sesbania with their rhizobia, some infection processes involved massive cell death -Capoen et al New phytol 2010 and Bonaldi et al 2011 MPMI-, which might involve NLR.
-15- L165 -167: about chitin: in addition to fungus the authors might consider mentioning insects, these organisms are chitin harboring organisms and many of them are detrimental for plants.
-16- 179-181: Perhaps the authors might be more specific: bacterial flagellins are not only recognized by FLS2 –flg22 is-, see Fliegman & Felix 2016 Nature Plants –there might be even more recent papers about this issue-.
-17- L199-202 I might have misunderstood as English is not my native language but I am surprised that “early” is used for a reference published in 2008 whereas lines 244-246 “current” is use for a paper published in 2007.
-18- About the positive role of ROS in rhizobia-legume interactions, in addition to papers dealing with RBOH, the authors might consider to cite the work published in J bact (Jamet et al 2007), the authors used a strain that overexpress a catalase to show the importance of ROS during the symbiotic process.
-19- Again about the role of nod factors in eliciting/preventing defense-like response, it would be relevant to discuss Anna Pietraszewska-Bogiel et al PLoSONE 2013 as well as Moling et al plant cell 2014.
-20- Line 283: Perhaps it is present elsewhere in the manuscript and I missed it but the authors might consider to provide a reference here for the statement “NFs induce specific defense gene upregulation via NFR”. As far as I remember there is no data supporting this idea in Liang et al.
-21- Line 386-389, perhaps it is worth mentioning that some nodulating rhizobia have no nod genes and no T3SS (Giraud et al science paper as well as Okazaki et al ISME 2016 doi: 10.1038/ismej.2015.103).
-22- Line 403 “via evolutionary mechanism or HGT”: HGT is an evolutionary mechanism.
-23- Line 443, in addition to ref 43, 81, 85, the authors might consider to cite here the reference 83 (Tang et al).
-24- The authors put emphasis on the role of NLR on calcium fluxes bit their role in NAD depletion might as well have dramatic influence on symbiosis; the authors might consider to equilibrate the 4.4 paragraph
-25- L571-573: To my knowledge, there is no evidence that NLRs limit plant colonization by parasitic rhizobia (in few studies they have been showed to prevent colonization by what would be mutualistic rhizobia…).
Reviewer 3 Report
The paper submitted for review is an excellent literature study based on a review of many scientific articles, most of which were published in the last 10 years.
The authors touch upon very important issues about the legumes and their symbiosis with Rhizobia.
The authors emphasize triggers on the legumes symbiosis with rhizobia via their innate immune system.
The issues discussed by the Authors are original.
They aptly suplement the information concerning symbiosis with Rhizobia described by other authors.
Table and figures are clear and understandable.
The conclusions of the article are consistent with the problems discussed.
The references are properly chosen and cited.
The paper needs some editorial corrections.
The review can be published in the International Journal of Molecular Sciences journal in present form.
Reviewer 4 Report
This review provides a special insight into plant immune response relevant for symbiotic interaction with rhizobia. It is easy to read and it contains comprehensive information on the subject. Some minor issues are unclear and minor text improvements are needed (see below).
I believe the review could gain clarity with a short introduction into the subject of flg22 epitope (first mentioned on L. 82 in the figure 1 legend and in the text on L. 184) and an information on which rhizobia possess it and which do not (L. 81-82: “However, many rhizobia flagellin lack the flg22 epitope for perception”).
It would also be nice to mention in which symbiotic plant-rhizobium pairs do the bacteria possess T3SS and in which they do not, do this have any impact on other relevant pathways?
Fig. 1: (i) NodD, NodABC (please capitalize for proteins), TtsI, MAP3K and MAP2K: shown on Fig. 1 but are not mentioned in the text or the figure legend. These items should be either walked through (with references) in the text/legend or omitted. (ii) “Rhizobia spp.” looks like a non-existing taxonomy term. (iii) The interesting regulatory connections described on Ll. 283-294 could be added to the figure.
L. 85: NF is spelled twice
L. 221: rhizobia
L. 241: Reference missing to support the statement “Conversely NFs have been shown to induce a small transient MTI response, similarly to MAMPs of plant-pathogen interactions”
L. 247 “NF and chitin receptor proteins” instead of “NF and chitin proteins”?
Ll. 366-376: the paragraph contains redundancy, the lines 367-371 seem out of place
L. 438: „innB“: innB gene or InnB protein?
L. 439: which species is USDA61?
L. 457: Transcriptomic analysis of which species?
L. 459: Bel2-5 of B. elkanii?
L 461: nfr1 mutant, the style suggests a bacterial mutant, but apparently a plant gene is meant
Ll. 465- 484: capitalize N in nop if talking about proteins (or use italic for genes), several occasions, also on L. 82
Ll. 482-483: “… TNLs have very different recognition and signalling mechanisms of recognising the same nop protein which …”: replace “of recognising” with “for”, add a comma after “protein”
L. 497: “Ca2+ is often observed downstream of NLR activation” is confusing, shouldn’t it be something like “Ca2+ accumulation is often observed as a result of NLR activation”?
Reviewer 5 Report
This is a timely, wide-ranging and authoritative review on an important subject, namely the immune response in legumes infected by symbiotic rhizobia. The authors have done a good job in covering a very extensive range of literature, and have presented a nice update by incorporating ETI- and MTI-related references. That being said, there are still a few “gaps” that should be covered in a review of the nodulation immune response, which I detail below:
First, I was surprised to see little or no mention of hyper-promiscuous legumes like Phaseolus vulgaris, or Lotus burtii – how does the immune system work (or not) in these cases? See recent work by Zarrabian et al. (2022). Also, you don’t mention that “cheater” bacteria can avoid the immune response and “piggyback” their way into nodules via the infection thread – see Zgadzaj et al. (2015).
Lines 98-108. It is a bit more subtle than that, at least in the more “advanced” symbioses – see Westhoek et al. (2021), in which pea plants have shown an ability to selectively sanction less efficient (but still effective) Rhizobium symbionts if presented with better-performing strains.
Lines 223-233. The papers by Kawaharada et al. (2017a, b) are of relevance here.
Lines 234-235. Rhizobial LPS is a very important factor in forming a successful interaction with both legumes and non-legumes e.g. Mitra et al. (2016) and refs therein. Perhaps a little more could be discussed here?
Lines 318-331. Worth mentioning here that chitinases are essential for forming an effective nodulating symbiosis in Lotus (Malolepszy et al. 2018).
Lines 366-376. Non-root hair infection appears to be the evolutionary default infection process that has been retained in some legumes, such as Lotus (Madsen et al. 2010; Montiel et al. 2021), but also in flooding-tolerant legumes like Sesbania in which dual root hair/”crack entry” mechanisms can operate depending on the environment and/or the rhizobial strain e.g. there is an interesting interaction reported to occur between the model symbiotically-competent/endophytic Agrobacterium strain IRBG74 (Mitra et al. 2016) and Lotus in which IRBG74 infects L. japonicus via crack entry whereas Mesorhizobium loti uses the standard root hair entry (Montiel et al. (2021). Please expand a little more about immune responses, crack entry, and different rhizobial strains.
References
Kawaharada et al. (2017a) Molecular Plant-Microbe Interactions 30: 194-204.
Kawaharada et al. (2017b) Nature Communications DOI: 10.1038/ncomms14534
Madsen et al. (2010) Nature Communications 1: (DOI: 10.1038/ncomms1009).
Malolepszy et al. (2018) eLIFE 7: e38874 doi: 10.7554/eLife.38874
Mitra et al. (2016) Journal of Experimental Botany 67: 5869-5884.
Montiel et al. (2021) Plant Physiology 185: 1131-1147.
Westhoek et al. (2021) Proceedings of the National Academy of Sciences 118: e2025760118 doi: 10.1073/pnas.2025760118
Zarrabian et al. (2022) Molecular Plant-Microbe Interactions
Zgadzaj et al. (2015) PLoS Genetics DOI:10.1371/journal.pgen.1005280
